# Development of Equipment for Ski Mountaineering, a New Olympic Event

Lorenzo Bortolan [1,2,*], Barbara Pellegrini [1,2], Nina Verdel [3], Hans-Christer Holmberg [4,5] and Matej Supej [3]

1   Department of Engineering for Innovation Medicine, University of Verona, 37129 Verona, Italy;
    barbara.pellegrini@univr.it
2   CeRiSM, Sport Mountain and Health Research Centre, University of Verona and Trento, 38068 Rovereto, Italy
3   Faculty of Sport, University of Ljubljana, Gortanova 22, 1000 Ljubljana, Slovenia;
    nina.verdel@fsp.uni-lj.si (N.V.); matej.supej@fsp.uni-lj.si (M.S.)
4   Department of Health Sciences, Luleå University of Technology, 97187 Luleå, Sweden;
    integrativephysiobiomech@gmail.com
5   School of Kinesiology, University of British Columbia, Vancouver, BC V6T 1Z1, Canada
*   Correspondence: lorenzo.bortolan@univr.it

**Featured Application: This perspective paper describes the most current trends in the development of ski mountaineering equipment, along with the relevant regulations of the International Ski Mountaineering Federation. It also considers the limitations with respect to both the equipment now used and current regulations, providing guidelines designed to eliminate these deficiencies. This overview should assist coaches in the optimal selection of equipment for their athletes, designers in developing better products, and researchers who strive to incorporate the implications of scientific findings into both the engineering of equipment and athletic performance.**

**Abstract:** Ski mountaineering, a new Olympic winter sport involving both climbing and descending snowy slopes, requires considerable physical and technical abilities, as well as highly specialized equipment. Herein, we briefly describe this equipment and its influence on performance and consider potential future advances. Skis, boots, and bindings must be light enough to facilitate climbing uphill (in which as much as 85% of the total racing time is spent) and, at the same time, provide stability and safety in often-challenging descents. A skier must be able to easily and rapidly attach and remove the adhesive skins under the skis that provide grip while skiing uphill. Poles and their baskets must be designed optimally to transfer propulsive force and help maintain balance. Despite the popularity of ski mountaineering, research on this sport is scarce, and we indicate a number of areas wherein improvements in equipment could potentially advance both performance and safety. Such advances must be based on a better understanding of the biomechanics of ski mountaineering, which could be obtained with novel sensor technology and can be best achieved via more extensive collaboration between researchers, skiers and their coaches, and manufacturers of ski mountaineering equipment.

**Keywords:** performance; ski touring; skiing; SkiMo; racing; skis; poles; skins; innovation



## 1. Introduction

Ski mountaineering (SkiMo), which involves climbing and descending snowy slopes on skis, has become so popular that the individual sprint and mixed-gender relay will be included in the Winter Olympics, starting with the Milano Cortina Games in 2026 [1]. The various formats for SkiMo races differ primarily with respect to the total vertical gain, total distance, and pattern of downhill/uphill sections. In all events, much of the racing time is spent climbing uphill, which explains why a major requirement of SkiMo equipment is lightness [2]. Indeed, the combined weight of a skier and his/her equipment is inversely correlated with performance [1]. At the same time, the skis, boots, and binding must be

sufficiently stiff to provide control and safety on downhill terrain, where the snow is often ungroomed/unprepared.

In preparation for uphill climbs, removable adhesive skins designed to prevent sliding backward are attached to the bottom of the skis, and the boots are allowed to pivot freely around the anterior binding, thereby promoting a more natural gait. To reduce friction and allow greater speed during descents, the skier removes these skins and, in addition, locks in the posterior bindings and cuffs of the ski boots for better control. During races, there are special transition areas where these changes are made as rapidly as possible between an ascent and descent or, in some cases, between skiing and running/walking uphill.

Ekström (1980) [3] described skiing as "a relationship between man, equipment and environment and all these factors should be adapted to each other to obtain an optimal result". In recent decades, revolutionary advances in SkiMo equipment have strived not only to enhance racing performance [4,5], but also to reduce the risk of injury [6,7]. In elite competitions, where a few seconds can separate victory from defeat, the optimal choice of equipment, both with respect to performance and rapid transition, is crucial. Clearly, understanding the interaction between an athlete and his/her equipment is imperative for allowing the skier to achieve his/her full potential. At the same time, to ensure both safety and fair play, all equipment that is used must comply with specific rules imposed by the Federation.

In connection with the development of equipment for different individual sports, biomechanical research can help identify variables that influence both performance and the risk of injury (e.g., skiing technique uphill and downhill, step length and frequency, and choice of trajectory on a particular slope under prevailing snow conditions) [8]. However, few scientific studies have examined the biomechanical parameters associated with SkiMo [9,10] and even fewer have been performed in collaboration with manufacturers of SkiMo equipment. The aim of the present perspective is to describe and discuss the major improvements in equipment and in our understanding of the biomechanics of competitive Ski mountaineering in recent decades, as well as to consider potential future developments.

In light of the limited extent to which scientific research has focused on the development of SkiMo equipment, we attempted to bridge this gap in knowledge both by sharing our own experience and knowledge in this area and by seeking the insights of leading experts in the field. To achieve the latter, we conducted open interviews (starting with eight standard questions) with four members of the coaching staff/ski technicians associated with the national SkiMo teams of Italy, Norway, Sweden, and Slovenia. In addition, when more detailed information was desired, we posed questions directly to leading manufacturers of SkiMo equipment. Ethical principles regarding the voluntary participation of the individuals interviewed, as well as confidentiality and anonymity, were strictly adhered to.

The aim of the present perspective is to describe and discuss the major improvements in equipment and in our understanding of the biomechanics of competitive SkiMo in recent decades, as well as to consider potential future developments.

## 2. Skis and Bindings

The skis used in SkiMo racing competitions vary with respect to length, width, side-cut radius, camber profile, weight, bending and torsional stiffness, edges, and base. Typically, elite SkiMo athletes possess two or three pairs of skis that they use for different disciplines and under different snow conditions, as well as to replace skis damaged during a race. For example, light skis are designed especially for vertical events (with no downhill sections), and skis with more pronounced side-cut radii are for sprinting (events that are primarily performed on prepared/groomed snow). In contrast, alpine ski racers use 10–15 different pairs to navigate 4 disciplines (downhill, super giant, giant slalom, and slalom), which involve turns with quite different radii (Table 1).

**Table 1.** Comparison of the equipment utilized in ski mountaineering, cross-country skiing, and alpine skiing.

| Equipment | Parameter | Ski Mountaineering | Cross-Country Skiing | | Alpine Skiing | | | |
|---|---|---|---|---|---|---|---|---|
| | | | Classical | Skating | DH | SG | GS | SL |
| **Skis** | Minimum length M/W (cm) | 160/150 | Height of the skier minus 10 cm | | 218/210 | 210/205 | 193/188 | 165/155 |
| | Minimum weight per pair M/W (gr) | 1560/1460 (skis + bindings) | 750/750 | | - | - | - | - |
| | Minimum radius (m) | - | - | | 50/50 | 45/40 | 30/30 | |
| | Width of the shoulder/waist/tail of a ski (mm) | Min: 80/60/70 | - | | Max: 95/65/- | Max: 95/65/- | Max: 103/65/- | Min: -/63/- |
| | Additional information | Metallic edge covering at least 90% of a ski's length. | No wedge shape. | | Maximum distance between running surface and boot: 50 mm. | | | |
| **Bindings** | Additional information | Must have system for both lateral and forward complete release. Ski brakes are compulsory. | Commercially available bindings are acceptable. | | Ski binding that releases the boot from the ski when the load exceeds pre-set values. A ski brake designed to slow down a ski after the release of the ski's binding. | | | |
| **Boots** | Minimum weight per pair M/W (gr) | 1000/900 | - | | - | | | |
| | Additional information | In case of bikini liners, only the shell must cover the ankles. Notched rubber soles are obligatory. | Commercially available models are acceptable. | | Maximum thickness of ski boot soles: 43 mm. | | | |
| **Poles** | Maximum length (cm) | - | Height of the skier | Height of the skier × 0.8 | - | | | |
| | Additional information | Maximum diameter: 25 mm. No metallic basket. | Two poles of equal length. Must not provide unnatural energy that favors push-off. Fixed height (not telescopic). | | Metal baskets are not permitted. | | | |
| **Removable skins** | Additional information | Must cover at least 40% of length of the contact between a ski and snow. | - | | - | | | |
| **Helmet** | Certified as one of the following: | UIAA106—EN1077 class B EN12492—EN1077 class B | - | | ATM204 EN1077 cl. A | | | ATM204 EN1077 cl. B |
| **Suit** | Additional information | Three layers that fit the competitor's upper body. Two long-legged layers. | - | | Permeability of 30 L/m$^2$/s. Conformity with FIS specification CS2015. An undergarment that cuts resistance: minimal cutting force of 100 N; minimal uncut length of 200 mm. Remark: In addition, alpine skiers utilize other protective gear such as back protectors and inflatable airbags, as well as protective guards for the arms and shins. | | | |
| **Pair of gloves** | Additional information | Compulsory. | - | | Recommended. Protective padding allowed. | | | |
| **Ski goggles** | Additional information | Recommended. | - | | Recommended. | | | |
| **Other compulsory equipment** | | Snow shovel, snow probe, survival blanket, backpack, and whistle. | - | | - | | | |

DH: downhill; SG: super-G; GS: giant slalom; SL: slalom; M: men; W: women. "-": indicates that the characteristic is not covered by specific regulations or is not applicable to the sport.

Cross-country (XC) skiers have 30–50 pairs of classical and skating skis with different stiffnesses and structures/grindings specialized for various snow conditions/courses/disciplines [11]. The much smaller number of skis used for SkiMo may reflect the fact that the relationship between glide and grip is less critical than in the case of XC skiing. The gliding phase in SkiMo is less pronounced due to the use of skins that create greater friction

between the skis and snow, the absence of a ski camber, and the terrain, which is often steeper and requires opposing the force of gravity.

Since most of the racing time is spent skiing uphill, lightness is favored to a certain extent ("fast and energy-efficient"), which has led the International Ski Mountaineering Federation (ISMF) to impose a minimum weight for a ski and its binding combined (780 and 730 g for men and women, respectively) [12] (Table 1). In comparison, an alpine ski with its plate and binding, which must remain stable when subjected to the high ground reaction forces associated with skiing/turning on harder snow/ice surfaces at higher speeds, weighs more than 3.5 kg. On the other hand, an XC ski, constructed primarily to enhance performance on flat and uphill terrain, weighs less than 0.5 kg.

Competitive male and female ski mountaineers utilize skis that are as short as possible (to minimize weight) within the ISMF regulations (160 and 150 cm, respectively) [12], with a minimum width of 80 mm at the front, 60 mm in the center (waist width), and 70 mm at the tail end. This produces a circular arc (i.e., "side-cut") that facilitates turning downhill. SkiMo skis are similar in length to alpine slalom skis (>165 and 155 cm, respectively) but shorter than the skis utilized in the giant slalom and speed disciplines [13]. They are also shorter than XC skis [14].

Manufacturers of SkiMo racing skis attempt to achieve a balance between durability, stiffness, and lightness by utilizing a multilayer composite structure with a wooden or honeycomb softcore that guarantees good flexibility and low weight. This core is encased in different layers of carbon, basalt fibers, and glass fibers. To minimize friction and promote smooth performance [15], the base is composed of highly hydrophobic polyethylene, which minimizes interaction with water droplets while resisting abrasion with snow crystals. In addition, to reduce friction and thereby improve gliding even more, the surface of the ski base is subjected to stone grinding to produce a microstructure that further decreases the interaction between the polyethylene and water [16,17].

The edge of a ski is crucial to performance, and the edges of SkiMo racing skis are required to be made of metal and cover at least 90% of their length. Sharpening the edges, together with appropriate tuning, improves grip and turning [18,19]. Meanwhile, the thicker edges of alpine skis can be prepared using machines designed for this purpose, and the thinner and, thereby, lighter edges of SkiMo skis are usually prepared with a machine only at the start of the season and are prepared manually thereafter throughout the rest of the season.

SkiMo skis have spring bindings at both the front and rear of the boot, which, as with alpine skis, lock the boot in. The major difference between SkiMo and alpine ski bindings is the ability of the former to unlock the rear of the boot from the ski, thereby allowing the heel to be raised and the boot to pivot around the toe piece of the bindings, resulting in a more natural gait when skiing uphill (Figure 1).

The minimalistic, low-weight (110–150 g) pin bindings used in SkiMo skis consist of only a few moving parts, together with a toe and heel piece (Figure 1). The two pins on the toe piece clamp into the boot from both sides, thereby establishing the axis of rotation of the foot relative to the toe. The ISMF rules [12] prescribe that the skier must be able to lock the front piece without the use of any tool (Table 1). The heel piece usually has two pins that insert into slots on the boot, thereby holding the heel in place for skiing downhill. To allow the skier to walk uphill, these pins can be retracted or rotated 90 degrees so that they cannot insert into the slots on the heel of the boot, allowing the boot to pivot around the toe piece.

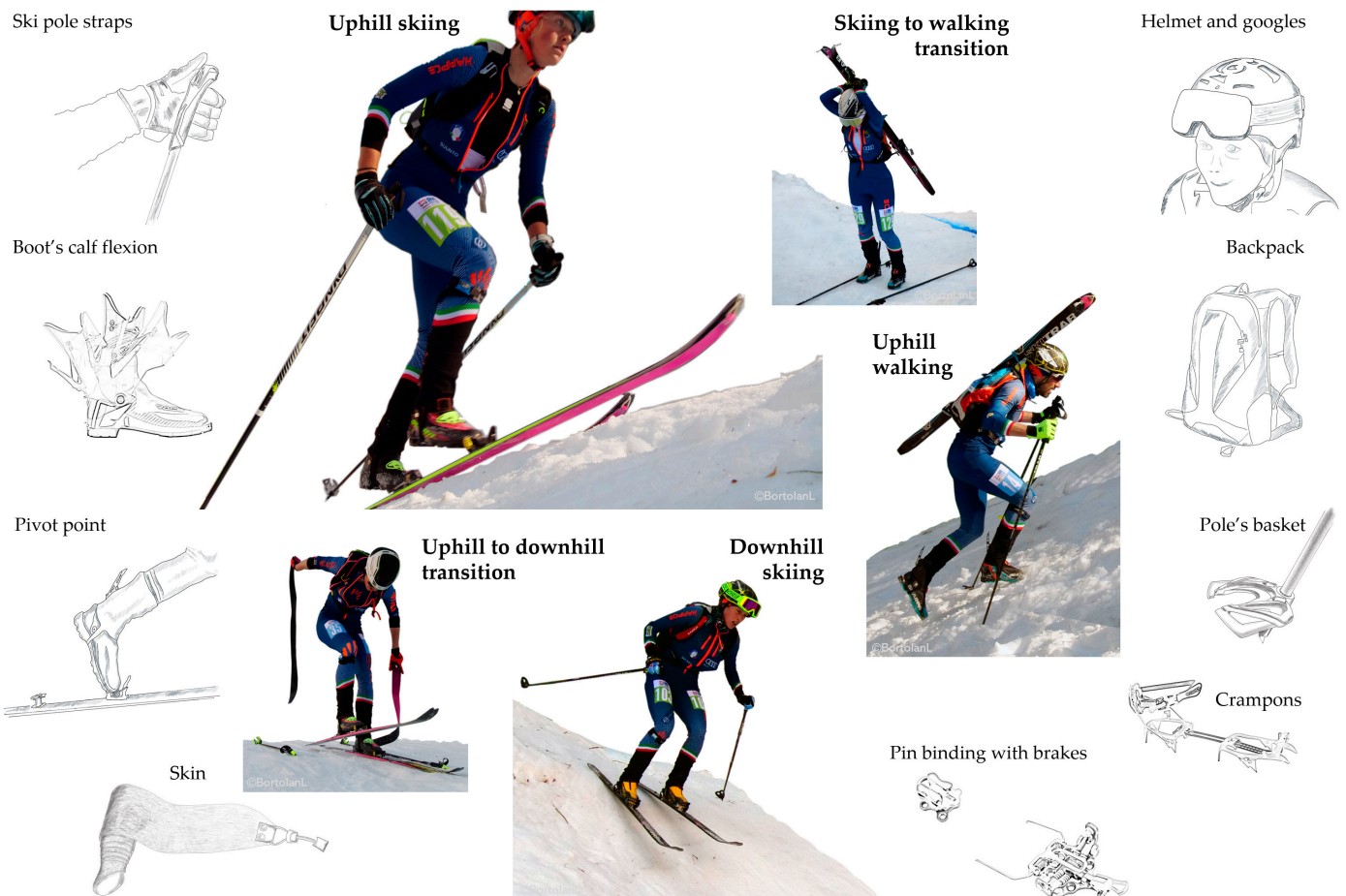

**Figure 1.** Major patterns in locomotion and transition phases (central part of the figure), and the equipment utilized during SkiMo competitions (photos: Lorenzo Bortolan).

In addition, the rear piece may also be adjusted to raise the heel to an extent determined by personal preference. This partially compensates for the inclination of the foot, thereby reducing the flexion of the ankle, lessening the strain on the calf, and helping to maintain a more upright posture when skiing/walking uphill [9,20]. While it is recommended that the height of the heel should be set in appropriate relation to the steepness of the slope [9,20], most SkiMo racing bindings have a fixed intermediate heel height of 20–38 mm (with a mean of 25 mm), depending on the manufacturer, for use on both flat and steep terrain.

Although, because of their lightness, pin bindings are most often used in connection with SkiMo, there are no standard requirements concerning the retention and release of these bindings, such as those that apply in connection with ski touring (ISO 13992:2014). Due to the possibility of inadvertent release by pin bindings [21], some skiers report that they lock the release mechanism of the toe binding during descent as well. For enhanced safety and reliability, these bindings should allow the adjustment of the clamping force and release load of the toe binding on the basis of the dynamics of the heel binding [22]. However, as updated in 2022, the ISMF rules prescribe the presence of a braking system to slow down the skis after the release of a binding. Unfortunately, although being of paramount importance for safety, binding release has not yet been investigated as extensively in the case of SkiMo as for other skiing disciplines [23–26].

Alpine skis have plates designed primarily to influence their bending and damping, as well as boot-out on steep inclinations [26]. However, because they are relatively heavy, such plates are not present in SkiMo skis.

### 3. Skins

To avoid slipping backward when skiing uphill, adhesive skins with combed hairs that grip the snow are applied to the base of SkiMo skis prior to an uphill section and then quickly removed in special transition areas before the subsequent descent (Figure 1). In this way, skins fulfill the same function as grip wax in connection with the diagonal stride technique in cross-country skiing, although skins provide even better grip and can, of course, be removed for skiing downhill. In the case of classical XC skiing, the interaction of the grip wax with the snow is influenced by the bending of the ski camber, which is, in turn, influenced by the load applied to the skis. Accordingly, the friction provided by the grip wax varies during the different phases of XC skiing, being higher during the kick and lower during the gliding phase [18,27,28]. Similarly, in SkiMo, the orientation of the hairs of the skins prevents the skis from slipping backward during the leg push while allowing forward motion during the gliding phase [29].

Frühwirth and colleagues [30] pulled a sled equipped with 12 different types of skins for 4 m at a constant speed of $0.36 \text{ ms}^{-1}$ and observed coefficients of gliding friction ranging from 0.22 for racing skins to 0.34 for those used in ski touring. Moreover, the coefficients of friction for these different skins varied in different ways with temperature. These gliding friction coefficients were 10-fold higher than that typical for XC skis [16] but only 10% of the grip friction when an apparatus was pulled with slowly increasing force in the grip direction of the skin.

Typically, SkiMo skiers utilize three or four different types of skins for different snow conditions (e.g., artificial versus fresh/new snow), many of which require some initial usage before obtaining their optimal glide and grip characteristics. In connection to certain teams and individual events, the ISMF regulations [12] allow the jury to request athletes to bring additional pairs of skins.

The difference between the gliding friction coefficient of XC skis and the grip friction of skins encourages technological developments that improve gliding via the latter while still maintaining adequate grip. The different attempts to reduce the friction of the skins utilized in SkiMo range from the usage of alternative materials (e.g., silicon rubbers) to skins that provide less grip and better glide specifically at the front of the skis, but these solutions have not yet been widely adopted by skiers themselves.

While the ISMF rules [12] state that these skins must cover at least 40% of the length of the contact between the skis and snow (Table 1), skiers usually attach a skin to the tip of a ski with a hook and extend it over the rear binding, which covers considerably more than 40%. In addition to its length, the extent of grip provided by a skin depends on the material from which it is made (e.g., mohair and/or nylon), the characteristics of the hairs (thickness, density, length, and inclination), and environmental conditions (temperature and humidity of the snow and air, as well as other characteristics of the snow).

Based on our interviews with the coaches and ski technicians associated with national SkiMo teams, another area that requires development involves the glues that adhere skins to skis. These must be strong enough to keep the skins in place during ascent and, at the same time, allow easy removal prior to descent. During a race, transitions must be as rapid and effective as possible, even when the skin and/or ski base are wet and/or snowy/frozen. To improve the balance between grip and glide even further, skins are also often waxed with a paraffin block containing compounds rich in fluorine in combination with liquid waxes that reduce the absorption of water, which could otherwise freeze under the skis.

When athletes are required to climb snow corridors or cross ridges with their skis attached to their backpacks, crampons, i.e., metal (usually aluminum) claw-like structures placed under their boots, may be used. Like mountaineering cleats, the sharp points on the side of a crampon are pushed into the snow while walking, which prevents sliding backward. Thus, these devices both facilitate climbing on hard crust or icy surfaces and improve safety.

## 4. Ski Boots

The major difference between the ski boots utilized in SkiMo and those used by alpine skiers is that the former are considerably lighter and designed so that the cuff can rotate around the ankle, allowing a switch to so-called "walking mode" and, at the same time, reducing the weight (Figure 1). Until minimum weight limits were set recently by the ISMF (Table 1), minimizing weight was the major challenge for manufacturers of SkiMo ski boots, and the relationship between the weight of a boot and the energy cost of skiing has been explored by a number of researchers [31–33]. Tosi reported a 2–3% increase in the energy cost when a 1 kg weight was attached to the ankle [25]. A more recent study on skiers moving at racing speed demonstrated that as little as 200 g of extra weight in the shoe increases the vertical energy cost by 3%, which can be detrimental to performance [34].

The development of new composite materials (e.g., based on personal communication with boots manufacturers molding technopolymers (Grilamid) reinforced with carbon fiber or full carbon) has now made it relatively easy to design extremely light ski boots that adhere to the ISMF weight limits. Therefore, the constructive challenge has now shifted towards the reliability of the equipment under different environmental conditions and on different types of terrain. When combining materials with different characteristics, the major difficulty lies in exploiting the advantages of each individual material without compromising those of the others [35]. To maximize safety, the ISMF states that boots must be designed to be used with metal crampons (Figure 1), and the entire sole must be covered with rubber at least 4 mm thick and with at least 8 indentations under the heel and 15 under the front section. This rubber surface facilitates walking/running on sections where the skis are carried on the backpack.

Another aspect to be taken into consideration is the ease with which the cuff moves with respect to the rest of the ski boot. Anecdotally, skiers prefer a boot with an extensive anteroposterior range of motion ("high flexibility") that allows a more efficient stride, both uphill and on flat terrain. On boots that are most admired, the cuff is connected to the remainder of the boot with ball bearings (instead of studs), which allows an extremely smooth and wide range of joint movement [36], while maintaining tight control of the supination and pronation of the feet. This is particularly important in the case of SkiMo because it makes it possible not only to climb directly up a maximal incline but also to ascend transversely in a zig-zag manner. When climbing transversely, the skis tend to follow the inclination of the slope, and the skier must exert pressure to avoid losing grip [37].

Competitive alpine ski boots are considerably stiffer than the boots utilized in SkiMo, wherein later models have begun to look more and more like XC ski skating boots. However, the former are lighter and lack a locking mechanism, which in SkiMo is obtained by using a single buckle (a quick, one-motion system positioned on the back between the cuff and the heel) that at the same time tightens the upper cuff and locks the flexion–extension at the ankle joint.

In summary, the varying mechanical properties of the different materials utilized, as well as their interactions, make SkiMo boots one of the most complex footwears used in winter sports. Future approaches, such as modeling, might improve the application of these properties and interactions to achieve even better reliability and safety [38].

## 5. Ski Poles

In addition to generating forward propulsion, which is their primary role in XC skiing, ski poles in SkiMo help maintain balance and facilitate coordination, both up- and downhill. In addition, poles may reduce the cost of vertical locomotion, as well as perceived exertion, as has been reported for trail-running uphill [39].

Aluminum, once the primary material from which SkiMo poles were composed, has now been replaced with carbon-fiber alloys and Kevlar wrapping. The rules concerning these poles [12] state only that their maximal permissible diameter is 25 mm and the basket should be composed of a non-metallic material (Table 1), imposing no other limits, such

as on pole length, as in the case of the FIS (Fédération Internationale de Ski) rules for XC skiing [11,40], or on the specific type of material used.

Although pole push is considered to be an important aspect of SkiMo, to date, the dynamics of this push or the relationship between the length of the pole and propulsive force remain to be examined. The length of SkiMo poles is intermediate between the lengths used by trail runners and by XC skiers for classical skiing on snow (the former being approximately 20–30 cm shorter than the latter). Indeed, the pole thrust associated with SkiMo appears to resemble that used during running or walking uphill on mountain paths [41,42], in contrast with its major propulsive role during XC skiing, which is most likely due to the steeper slopes and, consequently, slower speeds associated with SkiMo.

Unfortunately, there is little scientific research on this topic. According to the coaches/ski technicians interviewed, as well as our own experience, since most SkiMo is on un-groomed/unprepared snow, to prevent the tip of a pole from sinking too deep into the snow, the pole must be equipped with a wider ski pole basket (Figure 1). Moreover, usually, these baskets are asymmetrical in shape, which promotes the forward leaning of the shaft. Commonly, the baskets chosen are smaller when the snow is hard and slightly larger for new and/or soft snow. To date, no research on the effects of the geometry or size of the pole basket has been reported, but many SkiMo skiers prefer a relatively soft basket that more easily "molds itself to" the terrain and the surface of the snow. It is also important that the size and shape of the ski basket prevent it from contacting the ground before the pole tip, which could lead to loss of grip and slippage.

In contrast with XC ski poles, most SkiMo poles today do not have a strap into which the hand can be inserted, since this impedes the rapid release of the hand, as well as the rapid application and removal of skins, if the pole becomes stuck in the snow. However, some SkiMo skiers prefer to use quick-release straps, which attach the poles more effectively to the hand on one side but may slip away when pulled down on the ground during the transition phase (Figure 1).

Clearly, the influence of the poles on SkiMo performance requires a more detailed examination.

## 6. Upper- and Lower-Body Clothing, and Helmets

The ISMF regulations concerning SkiMo equipment include requirements for clothing that differ between the various types of events [12] (Table 1). Upper-body clothing, which should fit well, must be composed of three layers, including one long- or short-sleeved or sleeveless body-hugging layer; a ski suit or other second layer with long-sleeves; and a long-sleeved windbreaker jacket. Special pockets in upper-body clothing that can be zipped shut contain an avalanche transceiver and skins and gels. Lower-body clothing, which should also fit well, should be a ski suit or ski pants, together with windbreaker trousers that breathe.

Moreover, doubly certified helmets must be used, with chinstraps fastened. The effective ventilation of modern helmets allows them to be used with goggles (Figure 1). In addition, gloves that cover the entire hands up to the wrist should be worn throughout a race, as should a backpack that is sufficiently large to hold all of the equipment and with two straps for carrying the skis on its back and/or sides.

## 7. Concluding Remarks and Future Perspectives

The challenging environments in which elite SkiMo competitions take place require exceptional skills and well-designed equipment.

During a SkiMo race, as much as 85% of the total racing time is spent skiing up a considerable total elevation [2]. Accordingly, the combined weight of an athlete and his/her equipment exerts a profound impact on their performance [25]. Skis, bindings, and boots must be designed to optimize performance during ascents without compromising either performance or safety during the subsequent descent(s). Unfortunately, except for the weight of the boots, few features of SkiMo equipment and their relationship with performance have been investigated, which obviously prevents an in-depth discussion of

these matters. However, it is apparent that although the skins presently employed provide adequate grip, new insights gained from tribology should allow the next generation of skins to provide good grip in combination with improved glide. Until recently, research on the use of poles in SkiMo has suffered from low priority, but the increasing interest in this sport and, in particular, in shorter races that require delivering more power, should motivate more investigations on this subject in the future.

Furthermore, the ease and speed with which transitions can be performed during a SkiMo race are of key importance to the outcome, especially in connection with shorter races involving several transitions. For instance, transitions in the upcoming Olympic sprint event will take approximately 30% of the total racing time [43]. This aspect of SkiMo remains almost entirely unexplored, and biomechanical research and technological advances in this area have the potential to shorten the transition period significantly (always taking into consideration the ISMF regulations [12] in this context). For example, the development of "SMART" skis that adapt the characteristics of their base surfaces automatically to uphill or downhill skiing would completely eliminate the use of skins.

In this respect, analysis of the biomechanical aspects of SkiMo should provide numerous insights that, taking into consideration the characteristics of the individual skier (weight, sex, etc.), could help improve both performance and the design of equipment.

To date, most reports have focused on uphill skiing in the laboratory, and more measurements in the field (on snow) are now required. Such analysis would be facilitated by modern wearable technology, which enables more reliable and rapid collection and management of large datasets. As a result, important information relevant to performance can be provided more rapidly (perhaps in real-time) to skiers and their coaches, both during and following a race, as well as to the technicians involved in preparing the skis prior to the race.

Since present plans are to include only the sprint and mixed relays of SkiMo in the Olympic games, it can be expected that ski mountaineers who wish to participate will specialize in these disciplines, focusing on developing power, strength, and speed. At the same time, more equipment designed specifically for these disciplines is likely to become available. In general, further insight into this rapidly growing sport can best be achieved in collaboration with manufacturers of SkiMo equipment.

**Author Contributions:** Conceptualization and design, L.B., B.P., N.V., H.-C.H. and M.S.; literature search, L.B., N.V. and M.S.; writing—first draft preparation, L.B., B.P., N.V. and M.S.; writing sections of the manuscript, L.B., N.V., H.-C.H. and M.S. All authors contributed to the discussion section. All authors have read and agreed to the published version of the manuscript.

**Funding:** This research was financially supported by the Slovenian research agency ARRS (P5-0147).

**Institutional Review Board Statement:** Not applicable.

**Informed Consent Statement:** It was unnecessary to obtain consent from the skiers portrayed in the photographs since these were taken during an official competition where it was permissible to take and use photographs of the competitors.

**Data Availability Statement:** Not applicable.

**Conflicts of Interest:** The authors have no conflict of interest to declare.

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
