# Peer review of "Development of Equipment for Ski Mountaineering, a New Olympic Event"

_applsci, doi:10.3390/app13095339_

Round 1

Reviewer 1 Report

This perspective-manuscript aims to describe the equipment currently used by ski-mountaineers, its influence on performance and potential future advances. In general, I like this manuscript and it is well-written and nicely structured. However, I do have some questions and suggestions.

Comment 1: I do understand that this may be the first article regarding what kind of equipment most active ski-mountaineers use in competitions, and that is why this perspective is an interesting and important article. However, much of your statements are not referenced due to the low number of scientific studies in this sport. Therefore, I suppose that many of these statements are based on official regulations (i.e., ISMF) and your own experience and knowledge to this specific sport. Please correct me if I have misunderstood. However, I think this should be clearly addressed somewhere in this paper, although I don’t exactly know where to address it and how. It should maybe be clarified in the aim-section of the introduction or in the “Featured Application” paragraph at the beginning, that this is a description based on official regulations and the authors experience with the sport and athletes.   

Abstract

Line 26-28: This sentence is somewhat difficult to read. Can you divide this sentence into two?

Introduction

Line 41-48: This passage should be moved to the next paragraph and follow after the last sentence on line 55. I think this will provide a more natural build-up of the introduction.

Line 72-75: You state in the abstract that you wanted to “describe the equipment used, and its influence on performance and potential future advances”. However, in the main text you leave out the performance influence. Why is that? In my view, the performance part should be removed in the abstract. The paper does not address the direct impact of different equipment on performance, mostly due to the lack of scientific studies investigating this.

Skis and binding

Line 83-87: I suggest splitting this sentence into two.

Line 87: “The much small number…”. I suppose this should be “The much smaller number…”.

Line 98: “an XC ski”. Please change to “a XC ski”.

Concluding remarks and future perspectives

Line 313: Can you provide a reference for this statement?

Line 326: How do you know this?

General comment: I do follow your perspectives on future advances and development of equipment. However, I want to raise some reflections that may be beneficial for the paper to address. For instance, you state that the weight of the equipment (combined with the athlete’s weight) is crucial and that less is more. Can we be completely sure that lighter equipment always will be more beneficial for performance? Do we know how much a heavier ski boot will impact the overall performance compared to lighter ones? It may of course impact the ascent, but to what degree? A heavier boot may be more beneficial for stability and safety in the descent, which may be critical for maintaining a potential lead at the top. We clearly don’t have enough data on this. Although we anticipate the most logical way of development of equipment to be more beneficial, we must consider the possibility that more illogical ways of equipment development can turn out more beneficial for the overall performance. In addition, how will the development of equipment be influenced by the development of the sport in general (i.e., higher physical capacity, stronger athletes, etc.) If the sport develops into a closer race in the uphill-part, the downhill-part may be more important for the performance outcome. How can this influence the development of equipment? Since this paper show the insufficient number of scientific studies on these topics, I think it would be nice for the reader if you addressed these nuances and reflections as well.

Author Response

Thank you for the review and the suggestions that we tried to incorporate into the new version of the manuscript. The attached file includes the changes made in response to your comments.

Best regards.

Lorenzo bortolan

Reviewer 2 Report

Thank you very much for this interesting article. It contains a lot of important information. However, I have a couple of suggestions on how the article could possibly be of even greater use for reading. 

Lines 87-88: This explanation is not sophisticated enough for me. There may be other causes. If not, please substantiate this statement with a reference to the literature. 

Line 90-91. Can you please substantiate this statement. It is also a repetition of lines 51-52. These duplications should be avoided. 

Line 106: Please provide a source for these values. Thank you 

Line 115: Please provide a source. 

Line 157-199. please provide a source or more for the descriptions. 

Line 195-202. Please provide a source or more for the statements. 

Line 215: Please describe the differences in more detail and not just say that there are some. 

Line 217: Please give examples of new composite materials. 

Please add a table in which the differences to the other relevant ski disciplines regarding bindings, skis, loads, helmets, clothing etc. are made clearer than just in the body text. 

Lines 271 - 280: Please provide a source for the statement. 

Line 304: The heading is not appropriate. Concluding remarks should be much shorter. From line 305 onwards, there are then fewer conclusion for the future but rather numerous repetitions of already known statements (often without citing a source). 

Dear authors, 

This is an interesting article. Unfortunately, it still contains some aspects that should definitely be improved. For example, I almost completely miss the limitations of your statements or critical comments in the sense of a scientific discussion. 

Many, really many statements do not have a comprehensible source, as is usual in scientific journals. Please revise your contribution carefully.

Thank you. 

Author Response

Thank you for the review and the suggestions that we tried to incorporate into the new version of the manuscript. The attached file includes the changes made in response to your comments.

Best regards.

Lorenzo Bortolan

Round 2

Reviewer 1 Report

I thank the authors for responding to all questions and suggestions regarding this paper.

Reviewer 2 Report

Thank you for addressing all my comments. The manuscript has improved a lot and is in my opinion now ready for publication. Thank you so much.